

# The CCB-ID approach to tree species mapping with airborne imaging spectroscopy

Christopher B. Anderson[1,2]

[1] Department of Biology, Stanford University, Stanford, CA, USA
[2] Center for Conservation Biology, Stanford University, Stanford, CA, USA

Corresponding author
Christopher B. Anderson,
cbanders@stanford.edu

## ABSTRACT

**Background:** Biogeographers assess how species distributions and abundances affect the structure, function, and composition of ecosystems. Yet we face a major challenge: it is difficult to precisely map species across landscapes. Novel Earth observations could overcome this challenge for vegetation mapping. Airborne imaging spectrometers measure plant functional traits at high resolution, and these measurements can be used to identify tree species. In this paper, I describe a trait-based approach to species identification with imaging spectroscopy, the Center for Conservation Biology species identification (CCB-ID) method, which was developed as part of an ecological data science evaluation competition.

**Methods:** These methods were developed using airborne imaging spectroscopy data from the National Ecological Observatory Network (NEON). CCB-ID classified tree species using trait-based reflectance variation and decision tree-based machine learning models, approximating a morphological trait and dichotomous key method inspired by botanical classification. First, outliers were removed using a spectral variance threshold. The remaining samples were transformed using principal components analysis (PCA) and resampled to reduce common species biases. Gradient boosting and random forest classifiers were trained using the transformed and resampled feature data. Prediction probabilities were calibrated using sigmoid regression, and sample-scale predictions were averaged to the crown scale.

**Results:** CCB-ID received a rank-1 accuracy score of 0.919, and a cross-entropy cost score of 0.447 on the competition test data. Accuracy and specificity scores were high for all species, but precision and recall scores varied for rare species. PCA transformation improved accuracy scores compared to models trained using reflectance data, but outlier removal and data resampling exacerbated class imbalance problems.

**Discussion:** CCB-ID accurately classified tree species using NEON data, reporting the best scores among participants. However, it failed to overcome several species mapping challenges like precisely identifying rare species. Key takeaways include (1) selecting models using metrics beyond accuracy (e.g., recall) could improve rare species predictions, (2) within-genus trait variation may drive spectral separability, precluding efforts to distinguish between functionally convergent species, (3) outlier removal and data resampling can exacerbate class imbalance problems, and should be carefully implemented, (4) PCA transformation greatly improved model results, and (5) targeted feature selection could further

improve species classification models. CCB-ID is open source, designed for use with NEON data, and available to support species mapping efforts.

# INTRODUCTION

When you get down to it, biogeographers seek to answer two key questions: where are the species, and why are they where they are? Answering these simple questions has proven remarkably difficult. The former reflects a data gap; we do not have complete or unbiased information on where species occur. This is known as the "Wallacean shortfall" (*Whittaker et al., 2005*; *Bini et al., 2006*). Addressing the latter, however, does not necessarily require data; the drivers of species abundances and their spatial distributions can be derived from ecological theory (*McGill, 2010*). But evaluating these theoretical predictions does require data. Testing generalized theories of species distributions requires continuously-mapped presences and absences for many individuals across large areas. And while field efforts can assess fine-scale distribution patterns, they are often restricted to small areas. Mapping organism-scale species distributions over large landscapes could help fill the data gaps that preclude addressing these key biogeographic questions (*Anderson, 2018*). One remote sensing dataset holds the promise to do so for plants: airborne imaging spectroscopy.

Airborne imaging spectrometers measure variation in the biophysical properties of soils and vegetation at fine grain sizes across large areas (*Goetz et al., 1985*). In vegetation mapping, imaging spectroscopy can measure plant structural traits, like leaf area index and leaf angle distribution (*Broge & Leblanc, 2001*; *Asner & Martin, 2008*), and plant functional traits, like growth and defense compound concentrations (*Kokaly et al., 2009*; *Asner et al., 2015*). These traits tend to be highly conserved within tree species, and highly variable between species (i.e., interspecific trait variation is often much greater than intraspecific trait variation; *Townsend et al., 2007*; *Asner et al., 2011*). This trait conservation provides the conceptual and biophysical basis for species mapping with imaging spectroscopy. Indeed, airborne imaging spectroscopy has been used to map crown-scale species distributions across large extents in several contexts (*Fassnacht et al., 2016*). These approaches have been applied in temperate (*Baldeck et al., 2014*) and tropical ecosystems (*Hesketh & Sánchez-Azofeifa, 2012*), using multiple classification methods (*Feret & Asner, 2013*), and multiple sensors (*Clark, Roberts & Clark, 2005*; *Colgan et al., 2012*; *Baldeck et al., 2015*). However, this wide range of approaches has not yet identified a canonical best practice for tree species identification.

In this paper, I describe an approach to tree species classification using airborne imaging spectroscopy data that builds on the above methods and advances the discussion on best practices. This approach was developed as a submission to an Ecological Data

Science Evaluation competition (ECODSE; https://www.ecodse.org/) sponsored by the National Institute of Standards and Technology. This competition had participants use airborne imaging spectroscopy data, collected by the National Ecological Observatory Network's Airborne Observation Platform (NEON AOP; *Kampe et al., 2010*), to identify tree crowns to the species level. The work described was submitted to the ECODSE competition under the team name of the Stanford Center for Conservation Biology, and has since been formalized under the moniker Center for Conservation Biology species identification (CCB-ID) (https://github.com/stanford-ccb/ccb-id). First, I describe the CCB-ID approach to tree species classification using airborne imaging spectroscopy data. Next, I review its successes and shortcomings in the context of this competition. Finally, I highlight key opportunities to improve future imaging spectroscopy-based species classification approaches. The goals of this work are to improve NEON's operational tree species mapping capacity and to reduce barriers for addressing data gaps in plant biogeography.

## MATERIALS AND METHODS

CCB-ID was inspired by the botanical approache to species classification. In the field, botanists can use plant morphological features and a dichotomous key to identify tree species. These features often include variations in reproductive traits (e.g., flowering bodies, seeds), vascular traits (e.g., types of woody and non-woody tissue), and foliar traits (e.g., waxy or serrated leaves). The dichotomous key approach hierarchically partitions species until each can be identified using a specific combination of traits. Species classification with imaging spectroscopy is rather restricted in comparison; imaging spectrometers can only measure a subset of plant traits. This subset includes growth traits such as leaf chlorophyll and nitrogen content (*Lepine et al., 2016*), structural traits such as leaf cellulose and water content (*Papeş et al., 2010*), and defense traits such as leaf phenolic concentrations and lignin content (*McManus et al., 2016*). Furthermore, the inter and intraspecific variation in this subset of traits is rarely known a priori; this precludes the use of a standard dichotomous key (*Kichenin et al., 2013*; *Siefert et al., 2015*).

Classifying species with imaging spectroscopy instead relies on distinguishing species-specific variation in canopy reflectance. However, several confounding factors drive this variation including (1) measurement conditions (e.g., sun and sensor angles), (2) canopy structure (e.g., leaf area index or leaf angle distribution), (3) leaf morphology and physiology (i.e., plant functional traits), and (4) sensor noise (*Goetz et al., 1985*; *Ollinger, 2011*; *Lausch et al., 2016*). Measurement conditions and canopy structure tend to drive the majority of variation; up to 79–89% of spectral variance is driven by within-crown variation (*Baldeck & Asner, 2014*; *Yao et al., 2015*). Unfortunately, this variation does not help distinguish between species. Interspecific spectral variation is instead driven by functional trait variation (*Asner et al., 2011*; *Martin et al., 2018*). Disentangling trait-based variation from measurement and structure-based variation is thus central to mapping species with imaging spectroscopy.

CCB-ID classifies tree species using trait-based reflectance variation with decision tree-based machine learning models. This approach approximates a morphological trait and dichotomous key model to species mapping (*Godfray, 2007*), and is described in the following sections. The first section describes the outlier removal and data transformation procedures. The second section describes how the training data were resampled to reduce biases towards common species. The third section describes model selection, model training, and probability calibration. The fourth section describes the model performance metrics, and the final section describes two analyses performed post-ECODSE submission.

The NEON data provided in the ECODSE competition included the following products: (1) Woody plant vegetation structure (NEON.DP1.10098), (2) Spectrometer orthorectified surface directional reflectance–flightline (NEON.DP1.30008), (3) Ecosystem structure (NEON.DP3.30015), and (4) High-resolution orthorectified camera imagery (NEON. DP1.30010). These data were provided by the *ECODSE group (2017*; https://www.ecodse.org/) and are freely available from the NEON website (https://neonscience.org). These analyses used data product (2). All analyses were performed using the Python programming language (*Oliphant, 2007*; https://python.org) and the following open source packages: NumPy (*Van Der Walt, Colbert & Varoquaux, 2011*; http://numpy.org), scikit-learn (*Pedregosa et al., 2011*; http://scikit-learn.org), pandas (*McKinney, 2010*; https://pandas.pydata.org), and matplotlib (*Hunter, 2007*; https://matplotlib.org). The python scripts used for these analyses have been uploaded to a public GitHub repository (https://github.com/stanford-ccb/ccb-id), including a build script for a singularity container to ensure computational replicability (*Kurtzer, Sochat & Bauer, 2017*).

## Data preprocessing

The canopy reflectance data were preprocessed using two steps: outlier removal and dimensionality reduction. In the outlier removal step, the reflectance data were spectrally subset, transformed using principal components analysis (PCA), then thresholded to isolate spurious values. First, reflectance values from the blue region of the spectrum (0.38–0.49 μm) and from noisy bands (1.35–1.43 μm, 1.80–1.96 μm, and 2.48–2.51 μm) were removed. These bands correspond to wavelengths dominated by atmospheric water vapor, and do not track variations in plant traits (*Gao et al., 2009*; *Asner et al., 2015*). This reduced the data from 426 to 345 bands. Next, these spectrally-subset samples were transformed using PCA. The output components were whitened to zero mean and unit variance, and outliers were identified using a three-sigma threshold. Samples with values outside of +/− three standard deviations from the means (i.e., which did not fall within 99.7% of the variation for each component) for the first 20 principal components were excluded from analysis. These samples were expected to contain non-vegetation spectra (e.g., exposed soil), unusually bright or dark spectra, or anomalously noisy spectra (*Féret & Asner, 2014*). The outlier-removed reflectance profiles for each species are shown in Fig. 1.

Once the outliers were removed, the remaining spectra were transformed using PCA. This was not performed on the already-transformed data from the outlier removal process,
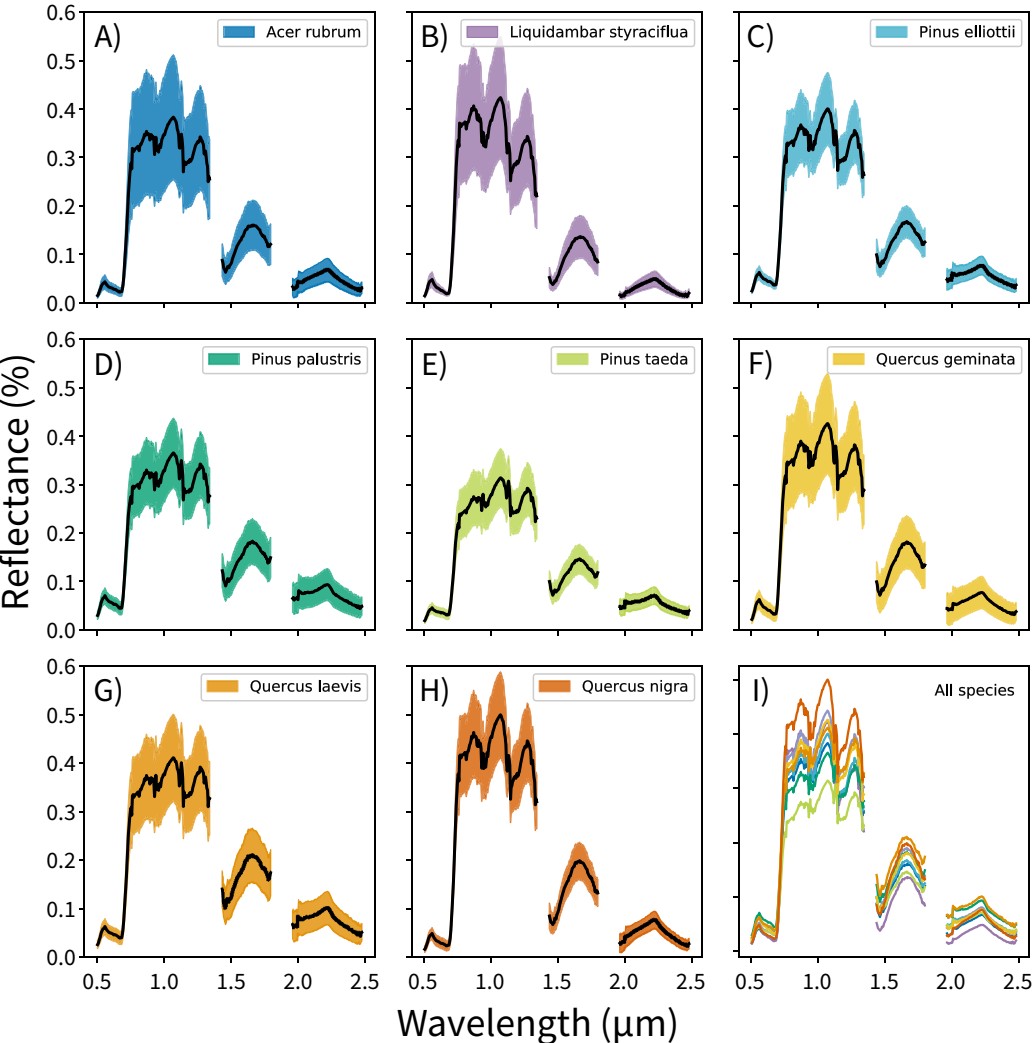

**Figure 1 Per-species canopy reflectance profiles.** (A–H) Canopy reflectance profiles for the eight tree species analyzed, with mean reflectance values in black and +/− 1 standard deviation values in color. (I) Mean reflectance values for all species, with each color corresponding to the individual species panels. Though the mean reflectance signals show high interspecific variation, the high intraspecific variation complicates classification efforts.

but on the outlier-removed, spectrally-subset reflectance data. PCA transformations are often applied to airborne imaging spectrometer data to handle the high degree of correlation between bands, and these transformations are highly sensitive to input feature variation (*Jia & Richards, 1999*). Furthermore, transforming reflectance data into principal components can isolate the variation driven by measurement conditions from variation driven by functional traits; this is critical for distinguishing between species. And though trait-based variation drives a small proportion of total reflectance signal, a single trait can be expressed in up to nine orthogonal components (*Asner et al., 2015*). After the transformation, the first 100 of 345 possible components were used as feature vectors for the species classification models. This threshold was arbitrary; it was set to capture the majority of biologically-relevant components and to exclude noisy components.

## Class imbalance

Class imbalance refers to datasets where the number of samples per class are not evenly distributed among classes. Imbalanced datasets are common in classification contexts, but can lead to problems if left unaddressed. Training classification models with imbalanced data can select for models that overpredict common classes when model performance is based on accuracy metrics. The ECODSE data were imbalanced: after outlier removal, these data contained a total of 6,034 samples from nine classes (eight identified species, one "other species" class). The most common species, *Pinus palustris*, contained 4,026 samples (66% of the samples) and the rarest species, *Liquidambar styraciflua*, contained 62 samples (1% of the samples).

These data were resampled prior to analysis to reduce the likelihood of overpredicting common species. Resampling was performed by setting a fixed number of samples per class, then undersampling or oversampling each class to that fixed number. This fixed number was set to 400 samples to split the difference of two orders of magnitude between the rarest and the most common classes. This number was arbitrary, but it approximates the number of per-species samples recommended in *Baldeck & Asner (2014)*. To create the final training data, classes with fewer than 400 samples were oversampled with replacement, and classes with more than 400 samples were undersampled without replacement. The final training data included 400 samples for each of the nine classes (3,600 samples total). Each sample contained a feature vector of the principal components derived from the outlier removed, spectrally subset canopy reflectance data.

## Model selection, training, and probability calibration

CCB-ID used two machine learning models: a gradient boosting classifier and a random forest classifier (*Friedman, 2001*; *Breiman, 2001*). These models can fit complex, non-linear relationships between response and feature data, can automatically handle interactions between features, and have built-in mechanisms to reduce overfitting (*Mascaro et al., 2014*). They were selected because they perform well in species mapping contexts (*Elith, Leathwick & Hastie, 2008*), in remote sensing contexts (*Pal, 2005*), and in conjunction with PCA transformations (*Rodríguez, Kuncheva & Alonso, 2006*). Furthermore, these models are built as ensembles of decision trees, resembling the dichotomous key employed by botanists. Unlike a dichotomous key, these models were trained to learn where to split the data since the trait variation that distinguishes species was not known a priori.

These models were fit using hyper-parameter tuning and probability calibration procedures. Model hyper-parameters were tuned by selecting the parameters that maximized mean F1 scores in fivefold cross-validation using an exhaustive grid search. F1 score calculates the weighted average of model precision and recall (see Model assessment), and maximizing F1 scores during model tuning reduces the likelihood of selecting hyper-parameters that overpredict common classes and underpredict rare classes. The following parameters were tuned for both models: number of estimators, maximum tree depth, minimum number of samples required to split a node, and minimum node impurity split threshold. The learning rate and node split quality criterion were also tuned for the gradient boosting and random forest classifiers,

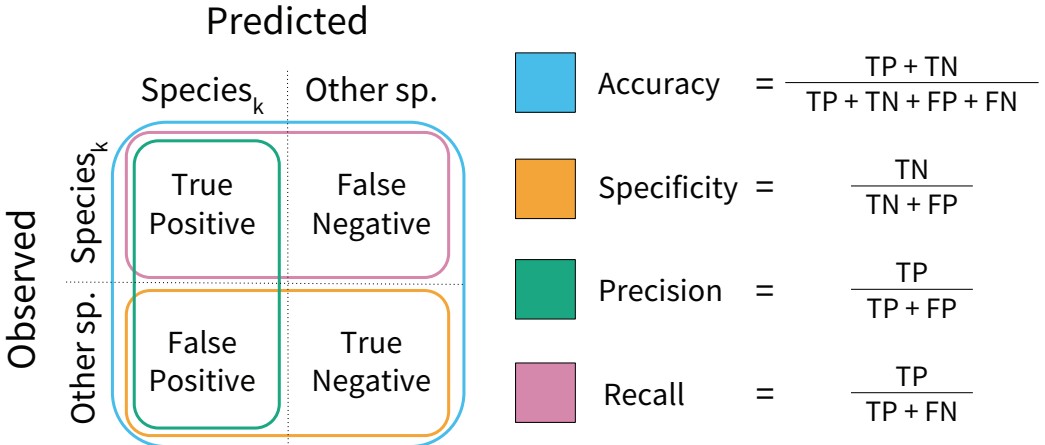

**Figure 2 Model performance metrics.** Visual representation of the classification model metrics calculated on a per-species basis. A confusion matrix was computed for each species, and each metric was calculated in a one-vs.-all fashion.      

respectively. All samples were used for hyper-parameter tuning, and the best model hyper-parameters (i.e., the hyper-parameters that maximized mean F1 scores in cross-validation) were used to fit the final models.

Accurately characterizing prediction probabilities is essential for error propagation and for assessing model reliability. Prediction probabilities were calibrated after the final hyper-parameters were selected. Well-calibrated probabilities should scale linearly with the true rate of misclassification (i.e., model predictions should not be under or overconfident). Some ensemble methods, such as random forest, tend to be poorly calibrated. Since ensemble methods average their predictions from a set of weak learners, which individually have high misclassification rates but gain predictive power post-ensemble, model variance can skew high probabilities away from one, and low probabilities away from zero. This results in sigmoid-shaped reliability diagrams (*DeGroot & Fienberg, 1983*; *Niculescu-Mizil & Caruana, 2005*).

To reduce these biases, prediction probabilities were calibrated using sigmoid regression for both the gradient boosting and random forest classifiers. The data were first randomly split into three subsets: model training (50%, or 200 samples per class), probability calibration training (25%, or 100 samples per class), and probability calibration testing (the remaining 25%). Each classifier was fit using the model training subset and the tuned hyper-parameters. Prediction probabilities were calibrated with sigmoid regression using the probability training subset and internal threefold cross-validation to assess the calibration. Calibrated model performance was assessed using the holdout test data. After these assessments, the final models were fit using the model training data, then calibrated using the full probability training and testing data (i.e., the full 50% of samples not used in initial model training).

## Model assessment

During model training, performance was assessed on a per-sample basis using model accuracy and log loss scores. Model accuracy calculates the proportion of correctly

classified samples in the test data (Fig. 2), and high model accuracy scores are desirable. Log loss assesses whether the prediction probabilities were well calibrated, penalizing incorrect and uncertain predictions. Low log loss scores indicate that misclassifications occur at rates close to the rates of predicted probabilities. During model testing, performance was assessed using rank-1 accuracy and cross-entropy cost (*Marconi et al., 2018*). Rank-1 accuracy was calculated based on which species ID was predicted with the highest probability. The cross-entropy score is similar to the log loss function, but was scaled using an indicator function. These can be interpreted in similar ways to accuracy and log loss; high rank-1 accuracy and low cross-entropy scores are desirable (*Hastie, Tibshirani & Friedman, 2009*).

Secondary model testing metrics were calculated for each species using the test data. These included model specificity, precision, and recall (Fig. 2). These metrics reveal model behavior that accuracy scores may obscure. Specificity assesses model performance on non-target species, penalizing overprediction of the target species (i.e., a high number of false positives). Precision also penalizes overprediction, but assesses the rate of overprediction relative to the rate of true positive predictions. Recall calculates the proportion of true positive predictions to the total number of positive observations per species. Higher values are desirable for each. These metrics were calculated to aid interpretation, but were not used to formally rank model performance.

Performance during model training was assessed at the sample scale, meaning the model performance metrics were calculated on every pixel (i.e., sample) in the training data. However, the competition evaluation metrics were calculated using crown-scale prediction probabilities, meaning the model performance metrics were calculated after aggregating each pixel from individual trees to unique crown identities. To address this scale mismatch, prediction probabilities were first calculated for each sample in a crown using both gradient boosting and random forest models. These sample-scale probabilities were then averaged by crown.

### Further analyses

Two post-submission analyses were performed to assess how PCA transformations affected model performance. Prior to these analyses, I bootstrapped the original model fits to assess their variance. I then compared these bootstrapped fits to models trained with the spectrally-subset reflectance data instead of the PCA transformed data. Next, I compared models trained using a varying number of principal components. These models were trained using $n_{\mathrm{pcs}} \in \{10, 20, \ldots, 345\}$ as the input features, with 345 being the maximum number of potential components after spectral subsetting. These comparisons assessed whether the PCA transformations improved model performance, and how changing the amount of spectral variation in the feature data affected performance. These analyses were each bootstrapped 50 times.

### RESULTS

CCB-ID performed well according to the ECODSE competition metrics, receiving a rank-1 accuracy score of 0.919, and a cross-entropy cost score of 0.447 on the test data. These were

**Table 1 Confusion matrix of classification results.**

| | | Predicted | | | | | | | | |
|---|---|---|---|---|---|---|---|---|---|---|
| | Species ID | *Acer rubrum* | *Liquidambar styraciflua* | *Other* | *Pinus elliottii* | *Pinus palustris* | *Pinus taeda* | *Quercus geminata* | *Quercus laevis* | *Quercus nigra* |
| Observed | *Acer rubrum* | **1** | 0 | 0 | 0 | 0 | 0 | 0 | 0 | 1 |
| | *Liquidambar styraciflua* | 0 | **1** | 0 | 0 | 0 | 0 | 0 | 0 | 0 |
| | Other | 1 | 1 | **1** | 0 | 0 | 0 | 0 | 0 | 0 |
| | *Pinus elliottii* | 0 | 0 | 0 | **0** | 1 | 1 | 0 | 0 | 0 |
| | *Pinus palustris* | 0 | 0 | 0 | 2 | **81** | 0 | 0 | 1 | 0 |
| | *Pinus taeda* | 0 | 0 | 1 | 0 | 0 | **4** | 1 | 0 | 0 |
| | *Quercus geminata* | 0 | 0 | 0 | 0 | 0 | 0 | **4** | 0 | 0 |
| | *Quercus laevis* | 0 | 0 | 0 | 0 | 1 | 0 | 0 | **22** | 0 |
| | *Quercus nigra* | 0 | 0 | 0 | 0 | 0 | 0 | 0 | 0 | **1** |

Notes:
Binary classification results of the CCB-ID model on the competition test data. These metrics were calculated using the independent crown data.
Bold entries highlight correct model predictions.

the highest rank-1 accuracy and the lowest cross-entropy cost scores among participants. Other methods reported rank-1 accuracy scores from 0.688 to 0.88 and cross-entropy scores from 0.877 to 1.448 (*Marconi et al., 2018*). A confusion matrix with the classification results is reported in Table 1. In addition to the high rank-1 accuracy and low cross-entropy cost scores, the CCB-ID model performed well according to the secondary crown-scale performance metrics. These secondary metrics calculated a mean accuracy score of 0.979, mean specificity score of 0.985, mean precision score of 0.614, and mean recall score of 0.713 across all species. The per-species secondary metrics are summarized in Fig. 3. These results were calculated using the categorical classification predictions (i.e., after assigning ones to the species with the highest predicted probability, and zeros to all other species). The probability-based confusion matrix and classification metrics are reported in Table S1 and Fig. S1, respectively.

During model training, outlier removal excluded 797 samples from analysis. A total of 264 of the 797 samples (33%) removed from analysis were from *P. palustris*, while the remaining 533 samples (67%) were from non-*P. palustris* species. Outlier removal disproportionately excluded samples from uncommon species; 45% of samples from *L. styraciflua,* the rarest species, were removed. After outlier removal, the first principal component contained 78% of the explained variance. However, this component did not drive model performance; it ranked 7th and 11th in terms of ranked feature importance scores for the gradient boosting and random forest classifiers. Model accuracy scores, calculated on a sample basis (i.e., not by crown) using the 25% training data holdout, were 0.933 for gradient boosting and 0.956 for random forest. Log loss scores, calculated prior to probability calibration, were 0.19 for gradient boosting, and 0.47 for random forest. After probability calibration, log loss scores were 0.24 for gradient boosting and 0.16 for random forest. The per-class secondary metrics reported a mean specificity score of 0.987, mean precision score of 0.908, and mean recall score of 0.907 across all species.
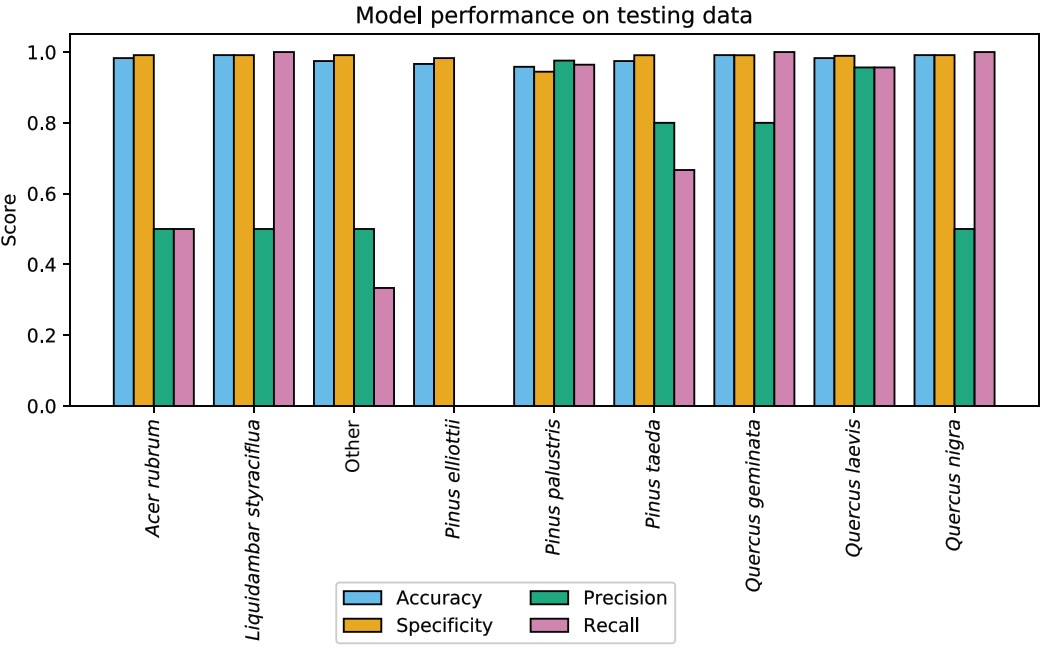

**Figure 3 CCB-ID model performance.** Per-species secondary performance metrics from the test data. These metrics were calculated using the binary confusion matrix reported in Table 1. Metrics weighted by the true negative rate (i.e., accuracy and specificity) were high for all species since the models correctly predicted the most common species, *Pinus palustris*. However, metrics weighted by the true positive rate (i.e., precision and recall) were much more variable since there were only one to six observed crowns for seven of the nine species (*P. palustris* and *Quercus laevis* had 84 and 23 crowns, respectively). This penalized misclassifications of rare species. These metrics were recalculated using the per-crown prediction probabilities, and can be found in Fig. S1.

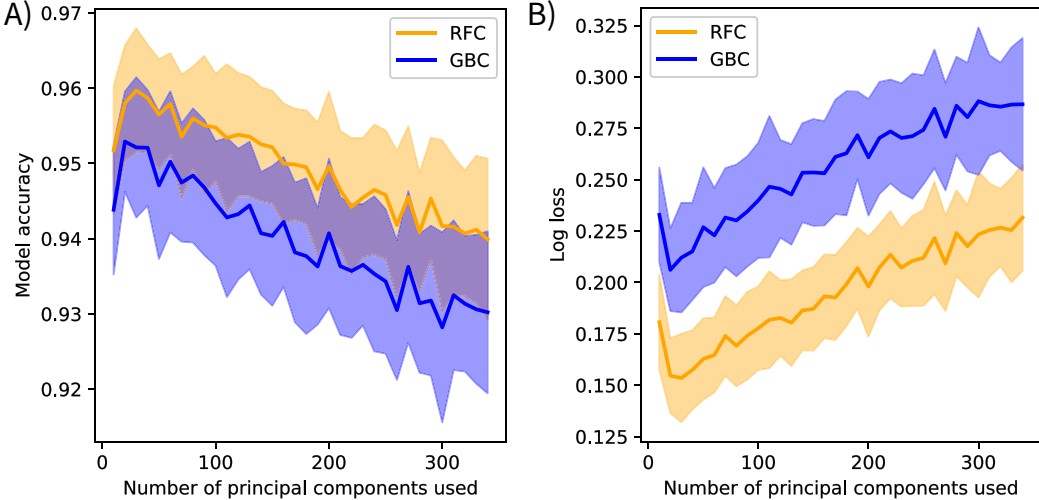

**Figure 4 Spectral variance and model performance.** The effects of increasing spectral variance on model performance through altering the number of principal component features. These plots show the mean (solid) and standard deviation (shaded) of (A) model accuracy and (B) log loss scores for each classification method. Scores were calculated on holdout data from the training set, not the competition test data. These results suggest that using all available spectral variance (i.e., all principal components) may decrease model performance. RFC stands for random forest classifier and GBC stands for gradient boosting classifier.

The post-submission analyses found PCA transformations improved model accuracy. Models fit using the original methods calculated mean bootstrapped accuracy scores of 0.944 ($s = 0.009$) for gradient boosting and 0.955 ($s = 0.008$) for random forest. Models fit using the spectrally-subset reflectance data as features calculated mean accuracy scores of 0.883 ($s = 0.012$) for gradient boosting and 0.877 ($s = 0.011$) for random forest, and mean log loss scores of 0.46 ($s = 0.03$) for gradient boosting and 0.48 ($s = 0.03$) for random forest. Mean model accuracies declined and mean log loss scores increased after including more than 20 components as features for the models fit using varying numbers of principal components (Fig. 4).

## DISCUSSION

CCB-ID accurately classified tree species using NEON imaging spectroscopy data, reporting the highest rank-1 accuracy score and lowest cross-entropy cost score among ECOSDE participants. These scores compare favorably to other imaging spectroscopy-based species classification efforts (*Fassnacht et al., 2016*). These crown-scale test results highlight the potential to develop species mapping methods that approximate botanical and taxonomic approaches to classification. However, this method failed to overcome several well-known species mapping challenges, like precisely identifying rare species. Below I discuss some key takeaways and suggest opportunities to improve future imaging spectroscopy-based species classification approaches.

The high per-species accuracy scores indicate a high proportion of correctly classified crowns in the test data. However, accuracy can be a misleading metric in imbalanced contexts. Since seven of the nine classes had six or fewer crowns in the test data (out of 126 total test crowns), classification metrics weighted by the true negative rate (i.e., accuracy and specificity) were expected to be high if the majority class were correctly predicted. Metrics weighted instead by the true positive rate (i.e., precision and recall) showed much higher variation across rare species, as a single misclassification greatly alters these metrics when there are few observed crowns (Fig. 3). Due to the small sample size, it is difficult to assess if these patterns portend problems at larger scales. For example, there were two observed *Acer rubrum* crowns in the test data, yet only one was correctly predicted. Was the misclassified crown an anomaly? Or will this low precision persist across the landscape, predicting *A. rubrum* occurrences at half its actual frequency? The latter seems unlikely, in this case; the low cross-entropy and log loss scores suggest misclassified crowns were appropriately uncertain in assigning the wrong label (Table S1). However, since airborne species mapping is employed to address large-scale ecological patterns where precision is key (e.g., in biogeography, macroecology, and biogeochemistry), we should be assessing classification performance on more than one or two crowns per species.

Comparing model performance between and within taxonomic groups revealed notable patterns. *Quercus* and *Pinus* individuals (i.e., Oaks and Pines) accounted for 120 of the 126 test crowns and there was high fidelity between them. Only one *Quercus* crown was misclassified as *Pinus*, and two *Pinus* crowns were misclassified as *Quercus*. From a botanical perspective, this makes sense; these genera exhibit very different growth

forms (i.e., different canopy structures and foliar traits), and should thus be easy to distinguish in reflectance data. However, within-genus model performance varied between *Quercus* and *Pinus. Quercus* crowns were never misclassified as other *Quercus* species, yet there were several within-*Pinus* misclassifications. This may be because *Quercus* species tightly conserve their canopy structures and foliar traits (*Cavender-Bares et al., 2016*), while *Pinus* species may express trait plasticity. *Pinus* species maintain similar growth forms (i.e., their needles grow in whorls bunched through the canopy), perhaps limiting opportunities to distinguish species-specific structural variation. Furthermore, they are distributed across the varying climates of the southern, eastern, and central United States, suggesting some degree of niche plasticity. If this plasticity is expressed in each species' functional traits, then convergence among species may then preclude trait-based classification efforts. Quantifying the extent to which foliar traits are conserved within and between species and genera will be essential for assessing the potential for imaging spectroscopy to map community composition across large extents (*Violle et al., 2012*; *Siefert et al., 2015*).

The post-submission analyses revealed further notable patterns. First, PCA transformation increased mean model accuracy scores compared to the spectrally-subset reflectance data. I suspect this is because the models could focus on the spectral variation driven by biologically meaningful components instead of searching for that signal in the reflectance spectrum where the majority of variation is driven by abiotic factors. The low feature importance scores of the first principal component support this interpretation. The first component in reflectance data is typically driven by brightness (i.e., not a driver of interspecific variation) and contained 78% of the explained reflectance variance, but ranked low in feature importance for both models. This preprocessing transformation approximates the "rotation forest" approach developed by *Rodríguez, Kuncheva & Alonso (2006)*, who found PCA preprocessing improved tree-based ensemble models in several contexts. They suggested retaining all components to maintain the original dimensionality of the input data. However, the analysis that varied the number of feature components showed model accuracy decreased when including more than the first 20 components (Fig. 4). This suggests that using all components overfits to noise. Performing feature selection on transformed data may help overcome this. Feature selection has been applied to reflectance data to find the spectral features that track functional trait variation (*Feilhauer, Asner & Martin, 2015*), and I believe it could help identify trait-based components that discriminate between species. Furthermore, other transformation methods may be more appropriate than PCA; principal components serve only as proxies for functional traits in this context. I expect transforming reflectance data directly into trait features, further extending the analogy of a taxonomic approach to classification, could improve species mapping efforts, improve model interpretability, and further develop the biophysical basis for species mapping with imaging spectroscopy.

Despite the successes of CCB-ID, there were several missteps in model design and implementation. For example, outlier removal and resampling were employed to reduce class imbalance problems but may instead have exacerbated them. First, the

PCA-based outlier removal excluded samples based on deviation from the mean of each component. However, since the transformations were calculated using imbalanced data, the majority of the variance was driven by variation in the most common class. This means outlier removal excluded samples that deviated too far from the mean-centered variance weighted by *P. palustris*. Indeed, 533 of the 797 samples excluded from analysis (67%) were from non-*P. palustris* species (which comprised only 37% of the full dataset). This removed up to 45% of samples from the rarest species (*L. styraciflua*), reducing the spectral variance these models should be trained to identify. This suggests outlier removal should either be skipped or implemented using other methods (e.g., using spectral mixture analysis to identify samples with high soil fractions) to reduce imbalance for rare species.

Data resampling further exacerbated class imbalance. By setting the resampling threshold an order of magnitude above the least sampled class, the rarest species were oversampled nearly 10-fold in model training. This oversampling inflated per-class model performance metrics by double-counting (or more) correctly classified samples for oversampled species. These metrics were further inflated as a result of how the train/test data were split. The split was performed after resampling, meaning the train/test data for oversampled species were likely not independent. This invalidated their use as true test data, overestimating performance during model training. This is unequivocally bad practice; I call this "user error." Undersampling the common species was also detrimental. Excluding samples from common species meant the models were exposed to less intraspecific spectral variation during training. This is a key source of variation the models should recognize. Excluding this spectral variation made it more difficult for the models to distinguish inter and intraspecific variation. Assigning sample weights (e.g., proportional to the number of samples per class) and using actually independent holdout data could overcome these issues. These will be implemented in future versions of CCB-ID. However, these need not be the only updates to this method; CCB-ID is an open source, freely available project (https://github.com/stanford-ccb/ccb-id). I invite you to use it and improve it.

## CONCLUSIONS

It was not always possible to classify tree species from airplanes; now it is. Airborne imaging spectrometers can map tree species at crown scales across large areas, and these data are now publicly available through NEON. However, there is currently no canonical imaging spectroscopy-based species mapping approach, limiting opportunities to explore key patterns in biogeography. The CCB-ID approach was developed to address this gap and to further the conversation on best practices for species mapping. CCB-ID performed well within the scope of the ECODSE competition, reporting the highest rank-1 accuracy and lowest cross-entropy scores among participants. Yet further testing is necessary to identify whether this method can scale to other regions (e.g., to high diversity forests). I hope CCB-ID will be used to improve future species mapping efforts to pursue answers to biogeography's great mysteries of where the species are, and why they are there.

## ACKNOWLEDGEMENTS

I would like to thank Gretchen Daily for her continued advisement, support, and inspiration. Thanks to the organizers of the NSF NEON workshop on mapping species, foliar chemistry and soil properties with spectroscopy, including Nancy Glenn, Nathan Leisso, Jessica Mitchell, Yi Qi, and Dar Roberts. Thanks to two anonymous reviewers for their insightful comments. Thanks to Phil Brodrick for being good at models, and even better at explaining them. Finally, thanks to Jeff Smith for comments on this manuscript, and for fruitful conversations on hyperspectral image mixing.

### Funding

C. B. Anderson was supported by the Bing-Mooney Fellowship in Environmental Science and Conservation at Stanford University's Department of Biology. The ECODSE competition was supported, in part, by a research grant from NIST IAD Data Science Research Program to D.Z. Wang, E.P. White, and S. Bohlman, by the Gordon and Betty Moore Foundation's Data-Driven Discovery Initiative through grant GBMF4563 to E.P. White, and by an NSF Dimension of Biodiversity program grant (DEB-1442280) to S. Bohlman. The National Ecological Observatory Network is a program sponsored by the National Science Foundation and operated under cooperative agreement by Battelle Memorial Institute. This material is based in part upon work supported by the National Science Foundation through the NEON Program. There was no additional external funding received for this study. The funders had no role in study design, data collection and analysis, decision to publish, or preparation of the manuscript.

### Grant Disclosures

The following grant information was disclosed by the authors:
Bing-Mooney Fellowship in Environmental Science and Conservation at Stanford University's Department of Biology.
NIST IAD Data Science Research Program.
Gordon and Betty Moore Foundation's Data-Driven Discovery Initiative through grant GBMF4563.
NSF Dimension of Biodiversity program grant (DEB-1442280).
National Ecological Observatory Network is a program sponsored by the National Science Foundation and operated under cooperative agreement by Battelle Memorial Institute.
National Science Foundation through the NEON Program.

### Competing Interests

The author declares that they have no competing interests.

### Author Contributions

- Christopher B. Anderson conceived and designed the experiments, performed the experiments, analyzed the data, contributed reagents/materials/analysis tools, prepared figures and/or tables, authored or reviewed drafts of the paper, approved the final draft.

## Data Availability

The raw data used in this analysis were provided by the "NIST DSE Plant Identification with NEON Remote Sensing Data" competition. These data are hosted on Zenodo: ECODSE group. (2017). ECODSE competition training set [Data set]. Zenodo. http://doi.org/10.5281/zenodo.1206101.

The code used to processes these data is the CCB-ID package. It is hosted on GitHub: https://github.com/stanford-ccb/ccb-id.

## Supplemental Information

Supplemental information for this article can be found online at http://dx.doi.org/10.7717/peerj.5666#supplemental-information.

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
