# Peer review of "The CCB-ID approach to tree species mapping with airborne imaging spectroscopy"

_PeerJ, doi:10.7717/peerj.5666_

## Round 0.1 · original submission · Minor Revisions

This is a well-written manuscript that requires only some revisions to the text. Please take care to add references to the text throughout to add context and demonstrate how this work builds upon the existing literature. Also pay attention in your revision to the stylistic, spelling, and acronym suggestions provided by the reviewers.

Reviewer 1 ·

Basic reporting

The manuscript is concise, clear, and includes a good background of advances and challenges of identifying tree species from hyperspectral images. The figures and tables are relevant and well-designed. The data used in this study are available from NEON. The author has provided the information necessary to locate and download the data.

Experimental design

The author introduced the CCB-ID approach to identifying trees, based on two machine learning methods (gradient boosting classifier and random forest classifier). Both methods were rigorously applied in terms of fitting parameters, training (model and probability calibration), and testing. The hyperspectral dataset was refined to remove spectral outliers and to reduce dimensionality (through PCA). The tree dataset was processed to remove class imbalance (some species with hundreds of samples, other species with tens or less samples). The author provided details of each step of data preprocessing and model fitting, training, and testing.

Validity of the findings

The CCB-ID approach proposed in this study produced promising results (high accuracy and low cross-entropy), but also identified limitations (mapping rare species) and components of the workflow that are problematic and need further development. The author summarized well the advantages of CCB-ID and the limitations and improvements needed.

Additional comments

I appreciate the author’s use of taxonomic approach to thinking about classifying trees from hyperspectral data. The methods are explained in detail. I think this study will inspire further work to fine-tune the CCB-ID approach. I enjoyed reading this manuscript. Below is a list of minor editing suggestions.
Abstract: spell out acronyms (CCB-ID, NIST, ECODSE, NEON); replace “obviate” with a more commonly used synonym – here and throughout the manuscript
Introduction: lines 34-35: spell out NIST and ECODSE
Materials and Methods
lines 77-78: mention that the data are freely available through NEON (in case ecodse.org will become inactive in a few years).
line 120: the specific epithet “styraciflua” is misspelled as “styaciflua”; the same epithet was misspelled as “stryaciflua” in Figures and Tables. Double check all scientific names.
line 193: the use of “sample scale” here is confusing; please explain at the beginning of Methods the difference between sample scale and crown-scale
Discussion
line 270: reword “showed different within…”
Conclusions: line 334: reword “wasn’t”

Reviewer 2 ·

Basic reporting

The manuscript has good voice and flow throughout. However, there are several opportunities to make more solid statements by adding references to them.

Experimental design

The author clearly explains the methods used, the assumptions made, and decisions to reach to the conclusions. However, there are some details about the research that only come later in the manuscript. The reader will benefit from having important details presented earlier in the manuscript, such as the sample sizes and how these changed throughout the different stages in the methods.

Validity of the findings

The author reports a new method to build on the efforts to classify tree species. However, the decisions made early in the methods could be influencing the reported overall accuracy and also be influencing further analyses. While the author acknowledges this situation, the way that overall findings are presented could be misleading.
I suggest to explicitly account for the uncertainty added by including a low sample size for underrepresented species in the overall accuracy assessment (e.g. report the accuracy of the method to discern a) all species and b) the species for which there are sufficient samples like Quercus and Pinus).

Additional comments

Line 2-14 This paragraph addresses the need for data to map species, and offer the possibility to do so via image spectroscopy. While this is true for vegetation, the text appear to be true for other groups. Later it is acknowledged, but I recommend to do so earlier.
Line 32 What NIST stands for?
Line 52 What are the characteristics of the referred subset? Are there any examples?
Lines 54-59 would benefit from a reference for the statement.
Lines 73 Are there important characteristics from the types of data used (e.g short description for each data set, sample size, etc.)
Line 109 Is there a reference range used elsewhere? E.g. why 100 and not 1000 ?, which is an exaggeration, but not knowing the typical range the reader cannot have an idea of the authors choice.
Line 146 What is F1?
Line 149 Likewise, what is RCF? Please indicate this and other acronyms
Line 211 How does this accuracy score compares to other participants? This will be useful to see the relevance of this metho
Line 259 With a sample size of 1, the result could be like tossing a coin. Please see the comment to Table 1
Line 301 Is there a reference that leads you to this inference?
Line 334 The use of contractions (i.e. Wasn’t ) was not used earlier, so probably avoid it here for consistence.
Table 1. I understand that the confusion matrix is at the sample level, which implies that the numbers in the tables correspond to the numbers of samples evaluated. If that is the case, why there is only 1 testing sample for several species? The author acknowledges this in the discussion but has not fully explain how this low sample size could be influencing the accuracies. These low samples should probably be excluded from the overall accuracy scores from the method used or at its best, provide the overall method accuracy with and without excluding these species samples.
Scaling –up. How is the data representative for other species/locations? This idea is barely touched in line 342 but not fully developed to see how it can scale-up.

---

## Round 0.2 · accepted · Accept

Thank you for addressing the reviewer's comments in detail. The paper is much improved and describes an interesting contribution to remote sensing science.

#